# Mechanistic Insights of Mitochondrial Dysfunction in Amyotrophic Lateral Sclerosis: An Update on a Lasting Relationship

**DOI:** 10.3390/metabo12030233

**Published:** 2022-03-09

**Authors:** Niccolò Candelise, Illari Salvatori, Silvia Scaricamazza, Valentina Nesci, Henri Zenuni, Alberto Ferri, Cristiana Valle

**Affiliations:** 1IRCCS Fondazione Santa Lucia, 00179 Rome, Italy; niccolo.candelise@gmail.com (N.C.); i.salvatori@hsantalucia.it (I.S.); silviascaricamazza@gmail.com (S.S.); valenesci@gmail.com (V.N.); 2Institute of Translational Pharmacology (IFT), Consiglio Nazionale delle Ricerche (CNR), 00133 Rome, Italy; 3Department of Experimental Medicine, Faculty of Medicine, University of Roma “La Sapienza”, 00185 Rome, Italy; 4Unit of Neurology, Department of Systems Medicine, University of Roma Tor Vergata, 00133 Rome, Italy; henri.zenuni@students.uniroma2.eu

**Keywords:** amyotrophic lateral sclerosis, motor neuron disease, mitochondria, metabolism, bioenergetic

## Abstract

Amyotrophic lateral sclerosis (ALS) is a fatal neurodegenerative disease characterized by progressive loss of the upper and lower motor neurons. Despite the increasing effort in understanding the etiopathology of ALS, it still remains an obscure disease, and no therapies are currently available to halt its progression. Following the discovery of the first gene associated with familial forms of ALS, Cu–Zn superoxide dismutase, it appeared evident that mitochondria were key elements in the onset of the pathology. However, as more and more ALS-related genes were discovered, the attention shifted from mitochondria impairment to other biological functions such as protein aggregation and RNA metabolism. In recent years, mitochondria have again earned central, mechanistic roles in the pathology, due to accumulating evidence of their derangement in ALS animal models and patients, often resulting in the dysregulation of the energetic metabolism. In this review, we first provide an update of the last lustrum on the molecular mechanisms by which the most well-known ALS-related proteins affect mitochondrial functions and cellular bioenergetics. Next, we focus on evidence gathered from human specimens and advance the concept of a cellular-specific mitochondrial “metabolic threshold”, which may appear pivotal in ALS pathogenesis.

## 1. Introduction

Amyotrophic lateral sclerosis (ALS) is a multisystemic, multifactorial neurodegenerative disease that mainly affects the upper and lower motor neurons (MNs) in the cortex, brainstem, and spinal cord [1,2]. The progressive loss of MNs causes muscle atrophy and gradual paralysis, leading to death usually occurring by respiratory failure. Similar to other neurodegenerative diseases, no therapies are currently available to halt the disease progression. With an incidence of 1.9 cases per 100,000 individuals per year, ALS is the most common motor neuron disease and the third neurodegenerative disease overall [1,2,3]. Although the pathophysiology and the etiology are still unknown, and most forms of ALS are sporadic (sALS), approximately 10% of cases have been linked to genetic mutations affecting a wide variety of genes [4], defining familial forms of ALS (fALS). The first mutated gene associated with the onset of fALS was *Sod1* [5], coding for the Cu–Zn superoxide dismutase 1 (SOD1) protein, found in one-fifth of all fALS cases. SOD1 is a cytosolic and mitochondrial protein that acts as a scavenger of reactive oxygen species (ROS), byproducts of the mitochondrial oxidative phosphorylation (OXPHOS) that converts nutrients into energy in the form of ATP [5]. Due to this cellular function, the pathobiology of ALS was originally associated with mitochondrial impairments. However, as more ALS-associated mutations were discovered, the focus shifted toward other functional biological processes.

Thus far, 26 different loci have been related to ALS pathogenesis, affecting several cellular mechanisms [6]. Most notably, hexanucleotide repeat expansion in the chromosome 9 open reading frame 72 (*c9orf72*) gene and mutations in genes coding for RNA-binding proteins TAR DNA-binding protein 43 (TDP-43) and fused in sarcoma (FUS) were identified as common causes of fALS [7,8,9], channeling research efforts toward the study of RNA metabolism and protein aggregation as main dysfunctions associated with the onset of the pathology. Together, the evidence converges on the multifactorial nature of ALS, with impairments in basic cellular mechanisms such as RNA regulation, protein homeostasis, cytoskeletal dynamics, mitochondrial functions, and energetic metabolism. Nevertheless, recent findings brought back the mitochondria as key players in the etiopathogenesis of ALS. Indeed, outside of their canonical role as energy producers, mitochondria are involved in other cellular functions compromised in ALS such as Ca^2+^ homeostasis, regulation of apoptosis, and protein quality control [10]. Moreover, data directly linking mitochondria dysfunction to the etiology of ALS emerged from the discovery of fALS associated mutations in the mitochondrial protein coiled-coil-helix domain-containing protein 10 (CHCHD10) [11]. Lastly, a recent study reported that 16 of 21 ALS-causative genes alter the integrity of mitochondria-associated membranes (MAMs), compromising mitochondrial functions [12]. Along with studies reporting early mitochondrial dysfunction in most models of fALS and in patients [13,14,15], the collective evidence strongly suggests the roles of mitochondria and energy metabolism to be pivotal in the development of ALS.

Here, we provide a review of the mechanisms by which mitochondria are impaired in ALS. First, we selected the most common fALS-associated proteins. For each of them, we lay out an update of the last five years on the studies of their involvement in mitochondrial function, focusing on the mechanisms that lead to mitochondrial impairment. In the next sections, we present evidence of mitochondrial metabolic alterations in ALS patients and broadly discuss how ALS phenotype may arise in the context of mitochondrial dysfunction related to bioenergetic and disease-modifying factors. 

## 2. Search Methods, Eligibility Criteria, and Screening

For the first section, the date of publication was restricted from 1 January 2016 to the present to cover a period of five years. As search terms, we used the target protein (e.g., “SOD1”) AND “mitochondria” AND “amyotrophic lateral sclerosis”. We further restricted to research papers by considering only “journal articles”.

The search, conducted on 1 November 2021, yielded 91 results for “SOD1”, 47 results for “TDP-43”, 23 results for “C9orf72”, 19 results for “FUS”, and 15 results for CHCHD10; the “other genes” section included 7 results for “TBK1”, 6 results for “OPTN” and 3 results for “Sigma-1R”. 

We screened these results to exclusively include research papers that provide novel mechanistic connections between the target protein and mitochondrial dysfunctions. The final groups included 7 results for SOD1, 10 results for TDP-43, 10 results for C9ORF72, 9 results for FUS, and 10 results for CHCHD10; the “other genes” section included 7 results for “TBK1”, 6 results for OPTN”, and 1 result for “Sigma-1R”.

To dissect the role of mitochondrial dysfunction in ALS, in the last following section, we focused on sporadic forms of ALS. Besides the aforementioned search criteria, we added the terms “human” AND “sporadic”. Moreover, instead of the target gene, we looked for experimental models (e.g., fibroblast, iPSC), obtaining 27 results for “motor neuron”, 3 results for “postmortem”, 11 results for “fibroblasts”, and 7 results for “iPSC”.

## 3. Involvement of fALS-Associated Proteins in Mitochondrial Dysfunction: An Update

In this section, we provide a literature review of the last five years on the connection between mitochondrial impairment and the most common proteins involved in ALS onset—namely, SOD1, TDP-43, C9ORF72, FUS, and CHCHD10, together with a grouped paragraph for the other genes involved (Table 1). We selected studies reporting a mechanistic explanation of how the target protein affects mitochondria, highlighting interactions among biomolecules and cellular pathways that ultimately lead to mitochondrial dysfunctions. 

### 3.1. SOD1

SOD1 is a ubiquitous homodimeric enzyme involved in the first antioxidant defense of the cell. The protein is an old acquaintance of the scientific community, as it was discovered and characterized several decades ago [72]. The original evidence of the genetic linkage between SOD1 mutations and ALS comes from a paper published in 1993 [5]. Since then, over 140 SOD1 mutations have been described, accounting for about 20% of fALS cases [73]. In parallel, an increasing variety of mutated SOD1 (mutSOD1) animal models, including rodents, Caenorhabditis, and zebrafish, as well as cell lines, were produced. Although alterations in mitochondrial metabolism, related to the expression of the mutated protein, were described, the mechanisms by which mutSOD1 affects mitochondrial metabolism remain quite obscure. Since the haploinsufficiency of the *sod1* gene is not an underlying mechanism of mutSOD1-associated ALS, the toxic activity of mutSOD1 has been linked to a gain of function of the protein [74]. SOD1 is present in both the cytosol and in the mitochondrial intermembrane space (IMS), where mutSOD1 was detected as proteinaceous aggregates in rodent models [16].

Although the contribution of the mitochondrial mutSOD1 aggregates to bioenergetic injuries is still under investigation, a growing body of evidence indicates the aberrant binding of mutSOD1 to voltage-dependent anion channel-1 (VDAC1) as a mechanism that could partially account for the energy deficit described in several SOD1-linked ALS models [75]. VDAC1 is an integral protein of the outer mitochondrial membrane (OMM) that controls metabolite flows, allowing for the transport of adenosine nucleotides and regulating cell processes such as apoptosis and Ca^2+^ homeostasis [76]. In two different ALS mouse models, mutSOD1 was found to specifically bind to the N-terminal domain of VDAC1, reducing its channel conductance [17].

Physiologically, VDAC1 regulates energy demand and apoptosis by binding to hexokinase1. This association couples glucose phosphorylation to mitochondrial production of ATP and blocks apoptosis by inhibition of Bax-induced release of cytochrome c [77]. MutSOD1, through the binding to VDAC1, was shown to displace hexokinase1, in turn promoting mitochondrial dysfunction and cell death [18]. Finally, as resident protein of MAMs, VDAC1 participates to form the IP3R–GRP75–VDAC complex, involved in the transport of Ca^2+^ from the endoplasmic reticulum (ER) to mitochondria [19]. For instance, in SOD1^G93A^ mice brain, a decrease in mitochondrial Ca^2+^ loading capacity was observed before disease onset, suggesting that loss of mitochondrial Ca^2+^ buffering may be causal in disease [20].

Therefore, it is conceivable to hypothesize that the mutSOD1-mediated inhibition of VDAC1 could alter mitochondrial Ca^2+^ buffering capacity, whose dysregulation has been widely described in several experimental models as well as in ALS patients [20,78]. MAMs are the physical connection between the OMM and subdomains of ER characterized by specific lipid and protein composition [79]. Since MAMs allow for a dynamic cross-talk between ER and mitochondria, they are involved in various cellular processes such as Ca^2+^ transport, lipid metabolism, autophagy, and insulin signaling [80]. Interestingly, mutSOD1 aggregates were found specifically in neuronal MAMs in the spinal cord of mutSOD1 mice [81]. The accumulation of mutSOD1 in MAMs was shown to disrupt their structure by inhibiting ER–mitochondria association, in turn inducing IP3R3 mislocalization, calpain activation, and mitochondrial dysfunction. Interestingly, studies reported that the pathological phenotype of mutSOD1 mice worsens when Sigma-1 receptor (Sigma-1R, detailed below) is genetically ablated, suggesting a functional interplay among mutSOD1, Sigma-1R, and MAMs [81]. Recently, a functional enrichment analysis obtained from a cross-ancestry genome-wide association study found a group of enriched genes involved in membrane trafficking, Golgi-to-ER trafficking, and autophagy within ALS-associated loci [82]. This result further tethers ALS to MAMs modulation of autophagic/mitophagic processes, already described with other ALS experimental paradigms [83].

Aside from VDAC1 and within the mitochondrial context, mutSOD1 aggregates were found to interact with the mitophagy receptor optineurin (OPTN) in neuronal cell lines, inhibiting its function as a promoter of the mitophagic flux. The sequestration of OPTN by mutSOD1 suggests that the accumulation of damaged mitochondria might contribute to the bioenergetic alterations described in ALS. Through an interactome analysis of misfolded SOD1 conformers in mitochondria from the ALS rat model, a recent paper identified the E3 ubiquitin ligase TNF receptor-associated factor 6, a multifunctional protein that plays a role in autophagy and mitophagy modulation [21], as a mutSOD1 interactor.

### 3.2. TDP-43

TDP-43 is a heterogeneous ribonuclear protein involved in mRNA biogenetic steps such as transcription, pre-mRNA splicing, nuclear export, and stability [84]. Mutations in *tardp* coding genes account for 4% of all fALS cases, and up to 97% of all ALS cases show TDP-43 positive inclusions in the motor cortex and in the spinal cord [85]. Although TDP-43 resides mainly in the nucleus in physiological conditions, it moves in the cytosol upon stress induction both in cellular and animal models of ALS [86,87,88]. TDP-43 cytosolic aggregates are phosphorylated, ubiquitinated, and cleaved in C-terminal fragments of 25 kDa and 35 kDa (CTF-25 and CTF-35, respectively).

Accumulating evidence shows a tight association between TDP-43 and mitochondria. In the brains of both patient and healthy cohorts, as well as in animal and cellular models, TDP-43 colocalizes with mitochondrial markers [89,90]. In the last five years, several lines of evidence reported a role for TDP-43 in noncanonical mitochondrial functions that could be grouped in three main classes: (i) interaction with either mitochondrial- or nuclear-encoded nucleic acids affecting mitochondrial functionality [22,23,24,25,26]; (ii) Ca^2+^ homeostasis, tethering TDP-43 and mitochondrial impairment to excitotoxicity [27]; (iii) cellular homeostasis and protein quality control, as TDP-43 was reported to interact with different proteins involved in mitochondrial unfolded protein response (UPR^mt^) and autophagy-related pathways [28,29,30,31]. Alteration in these functions ultimately affects mitochondrial inner membrane potential (m∆Φ), ATP synthesis, oxygen consumption rate, and/or ROS production, resulting in mitochondrial damage and energetic imbalance [28,29,30,31].

The interaction between TDP-43 and mitochondrial nucleic acids was first reported by Wang et al. (doi:10.1038/nm.4130.41), who showed that TDP-43 was shown to enter the mitochondria through three putative conserved sequences termed M1, M3, and M5, driving TDP-43 in the inner mitochondrial membrane (IMM) cristae. They further revealed that ablation of M1 is sufficient to block TDP-43 mitochondrial entry [91]. Similarly, we reported that the lack of M1 sequence in CTF-35 TDP-43 in neuronal cell cultures caused the accumulation of this fragment in IMS but not in IMM [92].

The interaction between TDP-43 and mitochondrial nucleic acids was first reported by Wang et al., demonstrating the binding between TDP-43 and mitochondrial RNA (mtRNA) coding for OXPHOS complex I subunits ND3 and ND6. This interaction affected the protein levels of ND3 and ND6 upon overexpression of TDP-43, in turn impairing complex I assembly and functionality causing the reduction in m∆Φ, ATP production, and oxygen consumption rate [91]. Similarly, Izumikawa et al. [23] showed the binding of TDP-43 to mtRNA. They showed that TDP-43 within mitochondria interacts through its RNA recognition motif 1 domain with UGUUU mtRNA stretches. Overexpression of TDP-43 resulted in the accumulation of unprocessed mtRNAs and decreased expression of target mitochondrial genes, including complex I subunits ND2 and ND5, thus affecting mitochondrial functionality. TDP-43 was further proposed to serve as a sponge for microRNA [24]. In cell lines, TDP-43 CTF-35 was shown to bind to a large subset of microRNA through its RNA recognition motif 1 domain. Notably, in this study, over 13% of upregulated genes induced by CTF-35 were nuclear DNA-encoded mitochondrial proteins, one-third of which were direct targets of TDP-43 trapped miRNA. Moreover, a set of mitochondrial genes were functionally sequestered by TDP-43 aggregation, causing a global mitochondrial imbalance, as mirrored by increased ROS production. The association of cytoplasmic TDP-43 with nuclear-encoded mitochondrial genes was further explored by Altman et al. [26]. Here, researchers took advantage of radial microfluidic chamber devices to dissect the spatial contribution of TDP-43 in different cellular models of the disease. They showed significant enrichment in TDP-43 bound RNA content in the axonal compartment. Next, they observed the suppression of local synthesis in the axonal and synaptic compartment by a click chemistry approach. Proteomic analyses revealed a global reduction in nuclear-encoded mitochondrial proteins, including respiratory chain complex proteins such as Cox4i1 and ATP5A1. Together, these results indicate that TDP-43 sequesters the RNA of nuclear-encoded mitochondrial genes in condensates, leading to the reduction in their local synthesis.

Overall, the evidence indicates that TDP-43 could affect mitochondrial functionality through multiple pathways. TDP-43 causative ALS mutations were further linked to the activation of the cGAS–STING inflammation pathway [25]. Briefly, researchers showed that through the mitochondrial permeability transition pore and VDAC1 activation, TDP-43 releases mitochondrial DNA (mtDNA) in the cytosol where it binds to cGAS. These findings reveal that the overexpression of TDP-43 causes the leakage of mtDNA into the cytosol, in turn activating the cGAS–STING inflammatory pathway.

Dafinca et al. reported a mechanistic association linking TDP-43 pathology to mitochondrial regulation of Ca^2+^ homeostasis [27]. Electrophysiological recordings on MNs derived from iPSCs from fALS patients harboring the TDP-43^M337V^ mutation showed increased recovery time upon exposure to glutamate, indicating a delay in Ca^2+^ buffering and increased vulnerability to glutamate excitotoxicity. RNA sequencing revealed an increase in the expression of AMPA- and NMDA-receptor subunits, as well as an upregulation of mitochondrial Ca^2+^ uptake regulator MICU1, although mitochondrial membrane potential showed no differences, compared with healthy control neurons. The evidence suggests that the mitochondrial Ca^2+^ buffering contributes to excitotoxicity as a critical pathomechanism in ALS.

Different lines of evidence put forward the central role of mitochondria in cellular homeostasis and protein quality control. Following the induction of overexpression of TDP-43 in cellular and animal models of ALS, Wang et al. [31] detected increased levels of mRNA coding for the UPR^mt^ components. These changes were associated with mitochondrial cristae damage, reduced activity of OXPHOS complexes I and IV, and decreased ATP synthesis. Among the altered transcripts, LonP1, an AAA+ ATPase involved in the degradation of misfolded and oxidized mitochondrial proteins, was confirmed as upregulated in human brain samples as well. Both wild-type and mutant TDP-43 bind to LonP1, whereas its downregulation led to increased mitochondrial localization of TDP-43, indicating that LonP1 directly degrades mitochondrial TDP-43 and protects neurons against TDP-43 induced toxicity [31]. Besides UPR^mt^, mitochondrial quality control is also regulated by autophagic mechanisms. Davis et al. [28] investigated mitophagy in HEK293 cells overexpressing or knocking down TDP-43 in wild-type mice cortex lysates. Immunoprecipitation experiments on mitochondrial fraction revealed an interaction between TDP-43 and Prohibitin 2 (PHB-2), a crucial receptor for Parkin-mediated mitophagy that interacts with LC3-II. TDP-43 overexpression reduced the amount of the autophagic marker LC3-II upon treatment with the autophagy inducer CCCP, suggesting the involvement of TDP-43 in the regulation of mitophagy. Parkin-mediated mitophagy was investigated by Jun et al. using cortical neurons overexpressing TDP-43 CTF-25 [29]. In this model, CTF-25 associates with Mitofusin-2 in damaged mitochondria and is related to the reduction in m∆Φ. Further, Parkin was shown to be recruited into the mitochondria, along with autophagosome markers LC3A/B and GABARAP-L1. Knockdown of mitophagy receptors TAX1BP, OPTN, or NDP52 resulted in an increased amount of insoluble CTF-25, suggesting that the fragment activates mitophagy and is directly degraded by this pathway. Researchers further explored the role of the motor protein Myosin IIB, showing the involvement of its ATPase activity for the association with mitochondria damaged by CTF-25 [29].

Lastly, a unique self-destructive pathway for mitochondria following overexpression of TDP-43, termed mitoautophagy, was recently described [30]. In mouse cortical neurons expressing ALS-associated TDP-43 mutations, mitochondria appeared to be elongating and folding, followed by the unification of the ends leading to disintegration of IMM and OMM. This pathway was not connected with the activation of the classical mitophagy pathway dependent on LC3B activation, suggesting an additional quality control system for mitochondrial welfare activated in response to TDP-43 pathology [30].

### 3.3. C9ORF72

Accounting for over 40% of fALS cases and up to 7% sALS, the repeat expansion of the noncoding hexanucleotide GGGGCC (G4C2) in intron 1 of the *c9orf72* gene [7,93], originally identified in 2011, is the most frequent genetic mutation associated with ALS. In contrast to healthy people, in whom G4C2 repeats are less than 30, in ALS patients with C9ORF72 mutation, these can reach up to thousands of copies. Nevertheless, in some cases, only 60 repeats have been observed to be sufficient to trigger the pathology [7,93]. Although the function of the protein is still poorly understood, bioinformatic analyses showed structural homology between C9ORF72 and DENN protein, a GDP/GTP exchange factor involved in membrane trafficking modulation [94]. Moreover, evidence suggests that C9ORF72 might be involved in autophagy regulation and in axonal and synaptic maintenance [95,96,97].

The toxicity of C9ORF72 in ALS is still debated. One hypothesis links C9ORF72 repeats to length-dependent RNA foci formation, causing RNA-binding protein sequestration, RNA processing machinery impairment [98], and blockade of translation following the formation of stress granules [99]. Another theory states that the reduction in the protein levels leads to endosomal trafficking inhibition and endocytosis perturbation, with autophagy as consequence [100]. Finally, it has been proposed that the repeat-associated non-ATG translation machinery generates dipeptide repeat proteins (DPRs)—specifically polyGA, polyGP, polyGR, polyPA, and polyPR—characterized by a propensity to aggregate in MNs [101]. Among them, polyGR and polyPR expression were shown to lead to nuclear stress [102,103], nuclear pore blocking [104], and DNA damage [105].

In 2016, Onesto et al. reported for the first time a correlation between C9ORF72-ALS mutation and mitochondrial dysfunctions [106]. By growing fibroblasts of C9ORF72-ALS patients in a galactose-rich medium, the authors observed mitochondrial morphological changes and functional defects such as membrane hyperpolarization and increased oxygen consumption, ROS, and ATP production. In the same cell model growth in a standard cell culture medium, Debska-Vielhaber et al. observed only a slight decrease in mitochondrial ATP production rates [107]. It is interesting to note that the use of a galactose-rich medium forces the transition from a glycolytic to oxidative metabolism, thus possibly justifying the differences in ATP production between the two studies.

Numerous other pieces of evidence have accumulated over the past five years, suggesting a direct action of C9ORF72 with mitochondrial functions. Reduced mitochondrial Ca^2+^ buffering capacity has been observed in iPSC-derived MNs of C9ORF72-ALS patients and is related to reduced expression of the regulatory protein MICU2 on the mitochondrial Ca^2+^ uniporter [32]. Furthermore, a phenotype of axonal homeostasis dysfunction related to a bioenergetic deficit was observed in the same cell model. This impairment seems to be due to defective mitochondrial respiration and reduced expression of the electron transport chain (ETC) genes of complex I and complex IV, encoded by mtDNA [33]. These mitochondrial alterations were also observed in the spinal MNs of the ventral horn but not in the sensory neurons of the dorsal horn of C9ORF72-ALS patients, revealing a selective vulnerability of the MNs toward C9ORF72 toxicity [33].

Moreover, several experimental lines of evidence suggest a toxic gain of function in DPRs. In particular, mitochondrial dysfunctions have been observed in cellular and animal models that overexpress polyGR. An interactome analysis conducted in human control neurons derived from iPSC ectopically expressing polyGR_(80)_ revealed the preferential binding of DPR to ribosomal proteins (67 out of 100 top interacting proteins). Two-thirds of these were mitochondrial ribosomal proteins mainly involved in the translation process of the subunits of the ETC complexes encoded by mtDNA [34]. Accordingly, an increase in membrane potential and oxidative stress has been observed in these cells and in iPSC-derived MNs of C9ORF72-ALS patients. Li et al. have reported that ∼ 60% of polyGRs localize within the mitochondria, where they are imported via the TOM complex. The Authors reported that the translation of this DPR occurs near the mitochondrial surface and that this process is often, and to date inexplicably, stalled. Consequently, this engorgement activates ribosome-associated quality control mechanisms and C-terminal extension, which, in turn, trigger the aggregation and toxicity of polyGRs in mitochondria [35]. Another study, in which an inducible mouse model of polyGR_(80)_ expression was used, suggested that this DPR preferentially binds to the ATP5A1 subunit of mitochondrial complex V and induces its degradation through ubiquitination [36]. Consistently, a reduced level of ATP5A1 protein expression was observed in mouse neurons expressing polyGR_(80)_ and in the brains of C9ORF72-ALS patients. The ectopic expression of ATP5A1 in polyGR_(80)_ neurons, as well as the reduction in polyGR expression during the symptomatic phase of the disease in mice models, yielded the recovery from the neurotoxic phenotype [36], thus linking the expression of polyGR to the progression of the disease.

Importantly, polyGR-mediated mitochondrial toxicity does not appear to be limited to neuronal districts. A recent study described abnormalities in the expression of mitochondrial genes, including the upregulation of RHOU and the downregulation of the ATP5A1 subunit of complex V, the NDUFB11 subunit of complex I, and the TIMM9 protein in C9ORF72-ALS myogenic progenitors and myocytes obtained from iPSCs of C9ORF72-ALS patients [37]. Mitochondrial defects have also been described in Drosophila muscles expressing polyGR. Specifically, it has been observed that polyGR directly binds to the mitochondrial contact site and cristae organizing system (MICOS), altering mitochondrial cristae morphology, ion homeostasis, and metabolism [38].

Finally, a recent study proposed that the mitochondrial toxicity induced by C9ORF72 may depend on haploinsufficiency, hence a loss of function due to the reduction in its protein expression. In this elegant study, Wang et al. described the mechanism of action of C9ORF72 at the mitochondrial level [39]. In healthy neurons, C9ORF72 was shown to be imported by the redox-sensitive AIFM1/CHCHD4 pathway into the IMS, where it allows the assembly of complex I by physically stabilizing the key assembly factor TIMMDC1 through the recruitment of PHB complex, which, in turn, inhibits AGF3L2-dependent TIMMDC1 degradation. Loss of function of C9ORF72 in different cell and animal models of haploinsufficiency, as well as in iPSC-derived MNs from C9ORF72 patients caused the failure to assemble mitochondrial complex I, leading to the imbalance of energy homeostasis.

### 3.4. FUS

FUS is an RNA-binding protein linked to oncogenesis and neurodegeneration [108] that was first related to ALS pathology in 2009 [9,109]. Akin to TDP-43, FUS shuttles between the nucleus, where exerts functions such as DNA repair, transcription [108,110], and splicing [111,112,113], and the cytosol, where it is involved in mRNA transport.

To date, over 30 mutations in the *fus* gene have been described in approximately 4% of fALS cases and in a small number of sALS cases. Deletions, truncations, and in-frame insertions in the *fus* gene have been identified mostly in the region coding for the C-terminal domain. Furthermore, mutations in the 3′UTR regulatory region, leading to wild-type FUS overexpression, have been identified in ALS patients [114,115].

The simultaneous reduction in FUS nuclear expression (loss of function) and accumulation in the cytosol (gain of function) have been proposed as leading pathogenic mechanisms in ALS [116]. The loss of FUS nuclear functions was shown to lead to neuronal cell death in Drosophila and zebrafish models [117], while cytoplasmic accumulation of FUS is often associated with the appearance of stress granules induced by various cellular stresses, including oxidative stress and mitochondrial dysfunctions [40]. Alterations in mitochondrial structure, mtDNA stability, and membrane potential have been correlated with FUS mutations [118]. Specifically, FUS was found to be involved in the repair of mtDNA damage, as it is required for PARP-1-dependent recruitment of the XRCC1/LigIII complex at oxidative DNA damage sites. Two familial FUS mutations were shown to impair mtDNA nick ligation: FUS^P525L^ causes mtDNA ligation defect due to the loss of functional FUS from the nucleus, whereas FUS^R521H/C^ fails to form the repair complex with XRCC1/LigIII and PARP-1, impeding its recruitment at the damage site [41,118].

FUS was reported to partially localize in mitochondria upon mutation or following cellular stress, leading to increases in the mitochondrial fission protein FIS1, mitochondrial fragmentation, ROS production, and loss of membrane potential [42,118]. The mitochondrial localization of both wild-type and mutated FUS was revealed to be dependent on the chaperonin HSP60. The downregulation of HSP60 reduced mitochondrial FUS levels and partially rescued FUS-induced phenotypes, including mitochondrial fragmentation and neurodegeneration [42]. Consistently, increased HSP60 expression was detected in brain tissue samples of FTD–FUS patients [42]. In rat primary neurons overexpressing FUS^R514G^, Salam et al. showed a decrease in colocalization between FUS and the mitochondrial anchor protein syntaphilin (SNPH) in the soma, mirrored by an increase in the movement of the mitochondria, resulting in altered synaptic functions, as mitochondrial motility strictly correlates with retrograde and anterograde synaptic trafficking [43].

Within mitochondria, FUS was shown to interact with the mitochondrial ATP synthase catalytic subunit ATP5B, interrupting the assembly of the ATP synthase complex and reducing mitochondrial ATP synthase activity, thus lowering mitochondrial ATP synthesis [44]. This scenario results in the accumulation of unassembled components of complex V, which leads to the activation of the UPR^mt^ response, worsening the mitochondrial damage. Indeed, the downregulation of UPR^mt^ or ATP5B subunit in FUS transgenic flies prevented mitochondrial damage and cell death [44]. Conversely, a recent study showed that mitochondria in MNs from FUS–ALS patients function properly, as the ATP-coupled oxygen consumption was not affected. However, the same study highlighted that the overexpression of mutant FUS^P525L^ affected the activity of dimerized ATP synthase complexes more than the overexpression of wild-type FUS. On the other hand, the reduction in mitochondrial ATP synthesis appeared similar upon overexpression of wild type or FUS^P525L^ [45].

In Drosophila muscle tissues, the expression of FUS caused a reduction in the levels and assembly of mitochondrial complex I and III subunits and a decrease in ATP production, although the expression levels of the complex V subunits of the ETC chain were unaltered. Parkin expression suppressed these mitochondrial dysfunctions, recovering the protein levels of mitochondrial complexes I and III, as well as the production of ATP [46]. However, previous studies reported contradictory results for Parkin activity in fly models of FUS [47].

FUS pathology was further mechanistically linked to the disruption of ER–mitochondrial associations. FUS was shown to activate GSK-3β, a regulator of the ER–mitochondria association, inhibiting the interaction between VAPB and PTPIP51 and in turn causing altered cellular homeostasis of Ca^2+^ and defective mitochondrial production of ATP. Accordingly, GSK-3β inhibition increases ER–mitochondria associations and recovers mitochondrial Ca^2+^ levels [48]. Nevertheless, Sakai et al. observed that the overexpression of FUS or VAPB resulted in increased or unchanged levels of MAMs, respectively [12].

Overall, a plethora of evidence suggests that FUS alters mitochondrial dynamics. Indeed, iPSC-derived MNs from ALS patients overexpressing FUS^R521H^ or FUS^P525L^ showed defects in axonal transport and ER–mitochondrial vesicle transport, as well as impaired MAMs. This phenotype was rescued either by the use of HDAC6 inhibitors or by its silencing via antisense oligonucleotides (ASOs), paving the way for novel therapeutic approaches [119].

### 3.5. CHCHD10

*Chchd10*, a member of the mitochondrial CHCHD protein family, is a novel and rare gene originally found in a French family to be causative of ALS [120,121].

Zhou et al. investigated the mutations in this gene in a cohort of 487 sALS and 12 fALS, reporting in the Chinese population a frequency of 0.4% within the sporadic subset [120]. The *chchd10* gene encodes for a small protein of 142 amino acids [121] situated in IMS [122,123]. CHCHD10 is highly expressed in cardiac and skeletal muscles, as well as in dopaminergic neurons of the midbrain and MNs in the spinal cord [11,124]. The protein structure consists of three domains: a positively charged N-terminal, which act as a mitochondrial targeting signal due to four interspaced arginine residues; a central hydrophobic domain; a C-terminal CHCH domain, which contains a CX(9)C (two pairs of cysteines each separated by nine residues) motif essential for mitochondrial protein import [11,121]. Although the specific function of CHCHD10 is still unclear, it is a known component of MICOS, a conserved protein complex involved in the formation, preservation, and stability of mitochondrial cristae. MICOS, composed of Mitofilin/Mic60 subunit and Mic10 subunit, plays a central role in maintaining contact sites between IMM and OMM, contributing to the formation of stable cristae junctions [49]. Mitochondrial cristae are propagations of the IMM that increase the available surface for the respiratory chain, hosting enzymatic complexes and making the OXPHOS more efficient. They are thus required to be functional for the correct localization of OXPHOS enzymes and for ATP production [50].

Mutations in CHCHD10 alter MICOS functionality, affecting the structural core of the protein complex. Indeed, CHCHD10 and its homologous CHCHD2 form heterodimers to maintain an efficient MICOS complex. CHCHD10–CHCHD2 complex is able to bind Mitofilin/Mic60, the central subunit of the MICOS core, promoting the disassembly of Mic60 and preventing the binding of other proteins, hence avoiding the formation of the functional MICOS complex. Several studies suggested a toxic gain of function of the CHCHD10–CHCHD2 heterodimer that results in loss of mitochondrial activities, as the heterodimer is deleterious for MICOS stability [51]. In the absence of MICOS, the activity of cytochrome-c oxidase (complex IV) appears reduced and the positioning of ETC complexes III and IV altered, leading to severe impairments in cellular respiration [52,53].

Another proposed pathological mechanism is the sequestration of CHCHD10 and CHCHD2 with other CX(9)C proteins essential for mitochondrial homeostasis. The mitochondrial disulfide relay protein Mia40 import system, which shuttles CHCHD10 in mitochondria, was proposed to be responsible for the sequestration of the protein in the IMM [54]. Mutations in the hydrophobic domain, such as the human CHCHD10 S59L, showed efficient import into mitochondria by Mia40 [122]. There, CHCHD10 and CHCHD2 aggregate in clusters leading to aberrant organelle morphology and function. These aggregates were shown to induce a strong mitochondrial integrated stress response (mtISR) through mTORC1 activation [55]. mtISR consists of the activation of stress-induced transcription factors and downregulation of respiratory chain enzymes. Physiologically, mtISR is an adaptive and protective process that spontaneously declines after transient mitochondrial insults, whereas a chronic mtISR may be unsafe, leading to cell and organ failure. Corey et al. generated a mutant mouse harboring the mutation S55L in CHCHD10, equivalent to the disease-associated human S59L mutation [56]. In transgenic mice, this mutation resulted in inappropriate mtISR. Accordingly, protein aggregation and misfolding in mitochondria are increasingly recognized as hallmark pathogenic events in mitochondrial disorders, characterized by loss of organelle structure and function [56].

Generally, most CHCHD10 patients show a slow disease progression. Nevertheless, a rapidly progressive ALS case was reported in a 29-year-old patient with the mutation Q108P in the CHCH domain [122]. Lehmer et al. highlighted that this mutation may inhibit mitochondrial import of CHCHD10 via the Mia40 system. The proposed mechanism suggested a reduced binding affinity to Mia40 caused by the Q108P mutation. Mia40 overexpression restores mitochondrial import [122], indicating that CHCHD10^Q108P^ mutation may lead to fast motor neuron degeneration due to a compromised mitochondrial import. Interestingly, this mechanism could explain the juvenile-onset and the more aggressive phenotype associated with Q108P mutation [122].

In the ALS–FTD spectrum, a co-interaction between CHCHD10 and TDP-43 was proposed [125]. The loss of CHCHD10 expression or activity appeared to drive TDP-43-induced deficits in mitochondrial fusion and respiration through the destruction of the optic atrophy 1 (OPA1)–mitofilin complex [57]. Mitofilin/Mic60 interacts with OPA1, an inner membrane GTPase that plays a role in mitochondrial fusion [58]. The interaction between OPA1 and mitofilin was shown to be essential for regulating IMM fusion and cristae integrity [58,59]. In 2020, Tian Liu et al. took advantage of the transgenic murine CHCHD10 variants R15L and S59L, showing that mutated CHCHD10 (S59L more strictly than R15L) destroys the OPA1–mitofilin complex in the brain and impairs mitochondrial fusion and respiration. On the other hand, wild-type CHCHD10 was reported to stimulate the association of this complex, indicating that CHCHD10 may be required for the interaction between OPA1 and mitofilin [57]. Although rare and still to be clarified, the CHCHD10 mutations in patients showed typical mitochondrial alterations—namely, abnormal mitochondrial cristae structure, deficiencies of respiratory chain complexes, and impaired mitochondrial respiration [126]. The discovery of CHCHD10 mutations in familial and sporadic cases of ALS/FTD–ALS reinforced the hypothesis that homeostasis of mitochondrial distribution is critical for the survival of cortical and spinal MNs [127].

### 3.6. Other Genes

Beyond the most common mutated genes linked to ALS, less common ones are also responsible for the alteration of mitochondrial functions. In the last five years, the mechanisms behind these dysfunctions were elucidated for only three of them: TANK-binding kinase 1 (*tbk1*), optineurin (*optn*), and sigma 1 receptor (*sigma-1R*). We grouped these “other genes” based on the mitochondrial mechanisms they affect: mitophagy (*tbk1* and *optn*) and Ca^2+^ homeostasis (*sigma-1R*).

#### 3.6.1. Mitophagy

Mitophagy is an autophagic process that selectively removes the damaged and dysfunctional mitochondria to maintain proper cellular functions [128,129,130]. Due to the importance of this pathway in neurons, defective mitophagy is frequently linked to neurodegenerative diseases. The first evidence of a connection between mitophagy and neurodegeneration occurred with the discovery of two mitophagy-linked proteins in Parkinson’s disease, PINK, and Parkin [131,132]. Mitophagy involvement was further observed in other neurodegenerative diseases, including ALS, in which the accumulation of damaged or dysfunctional mitochondria strongly contributed to the disease. Interestingly, mutations in two other proteins, TBK1 and OPTN, were shown to overlap both with ALS and defective mitophagy [133,134,135].

TBK1 is a multifunctional kinase, member of the IKK family, involved in the regulation of cellular pathways including immune response, cell proliferation, autophagy, and mitochondrial clearance [60,136,137]. Until now, more than 90 *tbk1* mutations have been identified in ALS, the majority of which are missense mutations with an unclear contribution to the pathology [133,134,138,139,140]. The latter kind of mutations affects kinase activity, dimerization, and autoactivation [61,62,63]. Particularly, mutations that reduce both TBK1 dimerization and kinase activity were indicated to strongly affect mitophagy, impeding the interaction with mitophagic adaptor OPTN [60,62]. Heterozygous nonsense and missense mutations in *optn* were identified in sALS and fALS patients, causing a reduced activity of the protein in the clearance of protein aggregates [141,142,143] and damaged mitochondria [60,64,144]. The ALS-linked mutation OPTN^E478G^ was demonstrated to be sufficient to disrupt mitophagy and to cause neurodegeneration in primary neurons [65]. The inability of OPTN mutants to bind ubiquitin and to associate with damaged mitochondria led to the failure in mitophagy initiation and thus to the accumulation of swollen mitochondria [65]. OPTN activity is regulated by TBK1, which phosphorylates OPTN at S177, in turn enhancing the binding of the protein to ubiquitinated substrates and promoting its translocation in damaged mitochondria [64,66]. Moore et al. found that TBK1 is recruited with OPTN to depolarize mitochondria for efficient mitophagy [67]. Researchers further observed that the expression of a mutant TBK1 unable to bind OPTN or the siRNA of endogenous OPTN prevented TBK1 recruitment to depolarized mitochondria. By contrast, OPTN was suggested to be required for TBK1 stabilization and activation to function as an autophagic receptor [67]. As these two proteins cooperate for mitochondrial clearance, a mutation in one partner may affect the other and vice versa, in either case leading to defective mitophagy.

#### 3.6.2. Ca^2+^ Homeostasis

Altered Ca^2+^ homeostasis is considered one of the main causes of motor neuron death in ALS. MNs are more vulnerable to Ca^2+^ dysregulation than the other cells, as they display a reduced ability to buffer cytosolic Ca^2+^ due to the high number of AMPA receptors in the postsynaptic terminal and low expression of Ca^2+^ buffering proteins such as parvalbumin and calbindin D28k. This background results in MNs developing a tight dependency on mitochondria for Ca^2+^ buffering [68,145,146,147,148]. In ALS, as mitochondria lose their functions and morphology, MNs fail to maintain functional Ca^2+^ homeostasis. Specifically, the major cause of altered Ca^2+^ homeostasis in ALS appears to depend on the loss of ER–mitochondria communication through MAMs [68]. As aforementioned, disruption of MAMs is shared by several ALS mutated genes such as SOD1, TDP43, FUS, and VAPB [69]. In recent years, the most studied gene correlated to ALS, MAM, and Ca^2+^ homeostasis has been the gene coding for Sigma-1R. This protein is an ER chaperone located in MAMs that controls Ca^2+^ signaling between ER and mitochondria through the binding with IP3R [68,69,70,71]. Recessive mutations in this gene were identified in juvenile forms of ALS. Sigma-1R mutants, including the newly discovered mutation p.L95fs, seem to be unstable and unable to bind IP3R. This loss of function of the protein resulted in the mislocalization of IP3R3 from MAMs and in the deregulation of Ca^2+^ homeostasis [81].

Together, the bodies of evidence gathered in the last five years converge on the central role of mitochondria in ALS pathophysiology. Multiple factors act on mitochondrial functions, involving different biological processes that ultimately result in mitochondrial alteration. Figure 1 provides a schematic view of the mitochondrial localization of fALS-associated proteins and the downstream pathways that lead to mitochondrial impairment.

## 4. Mitochondrial Alterations in Sporadic ALS Cases

Along with these data, evidence emerges of a direct role of different ALS-linked proteins in affecting mitochondria. Here, we review the advancement achieved in the last five years in understanding the link between mitochondrial dysfunction and the onset and progression of ALS. All the ALS-related genes are connected to mitochondrial functionality, causing their impairment and ultimately affecting energy production through a multitude of pathways. Several experimental lines of evidence describe that mitochondrial impairment is not only restricted to familial cases but also in sporadic ALS, where it has to be considered as a pathological hallmark (Table 2).

A postmortem study on the spinal cord of 23 sALS patients showed that complex IV activity was significantly decreased in the gray matter of both cervical and lumbar spinal cord [154]. A second postmortem study conducted on spinal cord gray matter of sALS patients identified 32 proteins that were up- or downregulated in the anterior horn. Some of them were involved in mitochondrial metabolism, OXPHOS, and glutathione homeostasis, whereas others were involved in Ca^2+^ homeostasis, protein metabolism, protein transport, and snRNP assembly [149]. A reduced expression of mtDNA respiratory genes was further found in sALS human cervical spinal cord and sALS MNs isolated by laser capture microdissection [152]. Veyrat-Durebex et al. showed that fibroblasts of sALS patients exhibit higher levels of mitochondrial heteroplasmy [153]. Accordingly, Singh et al. identified similar mitochondrial dysfunction in both sALS and fALS human iPSCs-derived MNs [155]. However, differences in mitochondrial pathophysiology between sALS and fALS have been reported in other iPSCs models [107,150,151]. Choi et al. investigated how the blockade of the PP1-DRP1 cascade effectively prevented mitochondrial defects and subsequent cell death in iPSC-derived human MNs [107]. Mitochondrial and metabolic studies on human peripheral blood mononuclear cells of sALS patients revealed several mitochondrial dysfunctions [20], although another study on fibroblasts of sALS patients found no differences in OXPHOS parameters [153].

## 5. The Threshold Effect of Mitochondrial Dysfunction

Existing data do not clarify whether the mitochondrial damage may be considered as the “primum movens” triggering the pathological phenotype in ALS. Notably, mitochondrial diseases are a group of genetic and chronic diseases, typical of childhood, generally characterized by severe course [156]. In ALS, it is conceivable to hypothesize subliminal mitochondrial damage without pathological significance in conditions of moderate energy demand. “Moderate” is, however, a relative concept, as the imbalance between energy demand and energy production may stem from different triggers. Hence, a “threshold” may be delineated, beyond which mitochondrial damage becomes evident and pathological.

During human aging, mitochondria undergo a process of senescence that compromises their function, often making them unable to satisfy even modest energy requirements [157,158,159]. This process, exacerbated in ALS, could be at the basis of the adult-onset of the disease. Specifically, given a permissive genetic and/or environmental background, the reduction in mitochondria functionality connected to normal aging may shift the threshold toward the onset of the symptoms.

In line with this hypothesis, the threshold could be overcome following increased physical activity. Although the correlation between ALS and physical activity has been widely debated since the 1990s, it has recently taken on a new prominence. After analyzing a large cohort of patients, Rosenbohm et al. reported that heavy occupational work intensity correlates with increased ALS risk, proposing that physical activity intensity may be a disease-modifying factor [160]. A recent study, using a Mendelian randomization approach and combining UK Biobank questionnaire items, indicated frequent and strenuous leisure-time physical activity as a risk factor for ALS, characterized by an increased penetrance in predisposing genotypic backgrounds. In particular, C9ORF72-patients presented a higher risk to develop exercise-induced ALS [161]. Moreover, a correlation between vigorous physical activity during early adulthood and adolescence and diagnosis of ALS before 60 years of age was recently reported [162].

As a whole, evidence emerged to support that strenuous physical activity might have a detrimental effect, whereas moderate exercise appears to be beneficial in ALS [163,164]. Using 31P magnetic resonance spectroscopy for the first time in vivo, Sassani et al. showed mitochondrial damage in the brain and skeletal muscle of ALS patients. Notably, the mitochondrial damage was amplified in patients undergoing physical exercise [165].

Broadly, mitochondrial dysfunction in ALS emerges when oxidative metabolism is challenged. Indeed, in several experimental paradigms of ALS cellular models, energy deficit appears only when cells are cultured in modified media that force the transition to oxidative metabolism, thus pushing cells to use mitochondria for the production of ATP [39,106,166,167,168,169].

## 6. Conclusions

In conclusion, given the high metabolic rate of the motor unit, it is plausible to hypothesize that ALS pathogenesis could partially originate from a failure in energy production as a response to coping with large energy expenditures. Indeed, the motor unit could be more likely to overcome the metabolic threshold. Hence, it is fathomable to envision that mitochondrial damage may be unbearable for some populations of cells, which may have a low “metabolic threshold”. The concerted effort of specific environmental and/or genetic clues, together with mitochondrial fatigue, would, in turn, justify the selective vulnerability of specific cells, promoting the onset of ALS phenotype.

Further studies are clearly required to better elucidate the central role of energetic metabolism in the etiopathogenesis and progression of ALS. Should this connection be unraveled, novel therapeutic approaches, based on personalized features such as environmental and genetic background, as well as lifestyle connected to energy expenditure, may arise to slow down or even halt the progression of the disease. In this regard, recent, promising evidence reported the application of drugs known to ameliorate mitochondrial function as candidates for this disease [170,171].

## Figures and Tables

**Figure 1 metabolites-12-00233-f001:**
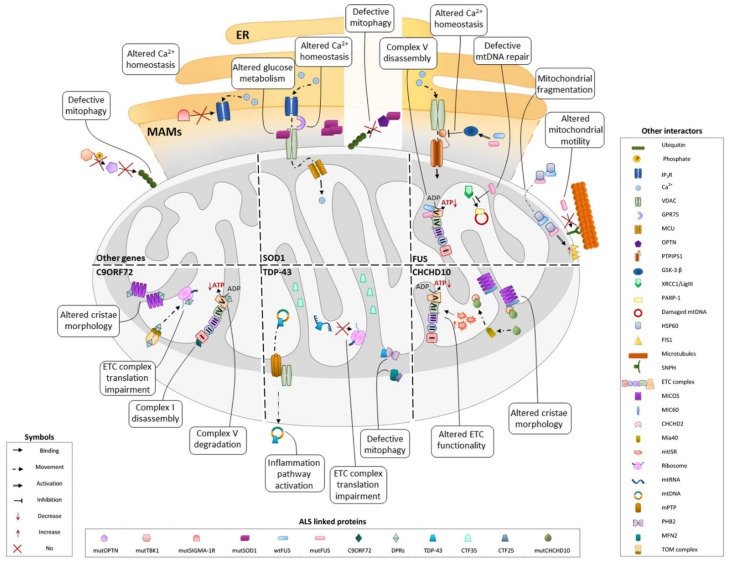
Latest mechanistic findings on principal ALS-related proteins in mitochondrial dysfunctions. Schematic representation showing a five years update of the mechanistic contribution of main ALS-related proteins in mitochondrial deficits.

**Table 1 metabolites-12-00233-t001:** Impact of ALS-related genes on mitochondrial functions. Pathogenic variants of ALS-proteins associated with altered mitochondrial functions.

Gene Symbol	Protein Symbol	Physiological Functions	Mitochondrial Related Dysfunction	Mitochondrial Localization	References
*sod1*	SOD1	Cytosolic antioxidant	ER-Ca^2+^ homeostasis Mitophagy Apoptosis	IMS OMM MAMs	[16,17,18,19,20,21]
*tardbp*	TDP-43	Splicing regulation RNA transport miRNA biogenesis Autophagy Stress response	ETC impairment Ca^2+^ homeostasis and excitotoxicity Mitophagy Inflammation mt genes regulation	IMS IMM cristae Matrix	[22,23,24,25,26,27,28,29,30,31]
*c9orf72*	C9ORF72	Transcription Splicing regulation Ribosome-associated quality control Endosomal trafficking Autophagy Axonal maintenance	ETC impairment Bioenergetic deficit Oxidative stress ER-Ca^2+^ homeostasis Morphology Quality control	IMM IMS cristae OMM proximity	[32,33,34,35,36,37,38,39]
*fus*	FUS	Splicing regulation RNA transport Maintenance of genomic integrity miRNA processing ER–mitochondria trafficking	ETC/oxidative stress ER-Ca^2+^ homeostasis Dynamics (axonal transport) mtDNA repair	IMM MAMs Matrix	[40,41,42,43,44,45,46,47,48]
*chchd10*	CHCHD10	MICOS integrity Oxidative phosphorylation	ETC impairment Structural integrity Dynamics (fusion/fission) Stress response	IMS IMM cristae	[49,50,51,52,53,54,55,56,57,58,59]
*tbk1*	TBK1	Autophagy Innate immunity signaling	Mitophagy	OMM proximity	[60,61,62,63]
*optn*	OPTN	Golgi maintenance and membrane trafficking Autophagy	Mitophagy	OMM proximity	[60,64,65,66,67]
*sigma1r*	SIGMA1R	ER–mitochondria trafficking Antioxidant response	ER-Ca^2+^ homeostasis	MAMs	[68,69,70,71]

Abbreviations: MAMs, mitochondria-associated membrane; IMS, mitochondrial intermembrane space; OMM, outer mitochondrial membrane; IMM, inner mitochondrial membrane; ER, endoplasmic reticulum; ETC, electron transport chain.

**Table 2 metabolites-12-00233-t002:** Main mitochondrial dysfunctions observed in sALS specimen and ex vivo models.

Altered Mitochondrial Function(s)	Experimental Model(s)	References
Respiratory chain and mitochondrial bioenergetic	Spinal cord sections	[149]
	iPSCs-derived MNs	[107,150]
	Fibroblasts	[107]
	PBMCs	[20]
Oxidative stress	iPSCs-derived MNs	[151]
	Fibroblasts	[107,150]
	PBMCs	[20]
Ca^2+^ homeostasis	Fibroblasts	[107,150]
	PBMCs	[20]
Mitochondrial distribution	Spinal cord sections	[107,150]
Mitochondrial biogenesis	PBMCs	[20]
mtDNA expression and protein involved in mitochondrial function	Spinal cord sections and laser-captured motor neurons	[152]
	Fibroblasts	[153]
	iPSCs-derived MNs	[151]

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
