# Peer review of "Mechanistic Insights of Mitochondrial Dysfunction in Amyotrophic Lateral Sclerosis: An Update on a Lasting Relationship"

_metabolites, 2022, doi:10.3390/metabo12030233_

Round 1

Reviewer 1 Report

Amyotrophic lateral sclerosis (ALS) is a severe neurodegenerative disease accompanied by the death of central and peripheral motor neurons, characterized by a progressive course.  Approximately 90% of cases are sporadic, with an unknown genetic component. Familial forms of ALS, comprising ~10% of cases, may be due to autosomal dominant or recessive mutations in various proteins.  At the moment, there are several theories of the pathogenesis of ALS. One of which, "mitochondrial", postulates that mitochondria are the earliest target in the pathogenesis of ALS, and their dysfunction affects the progression of the disease. The authors seem to share this point of view and present research data in this area over the past 5 years. The review considers both familial and sporadic forms of ALS. The material well presented and detailed and the introduction and discussion well referenced and reasonable. My suggestions for improvement would be minor textual modifications as follows:

  • Introduction

In the final paragraph of the section, the authors incorrectly indicate the order of presentation of the material. The authors state that "In the first section, we selected the most common fALS-associated proteins". But in the manuscript it is the third section. And then "In the second section, we present evidence of mitochondrial metabolic alterations..." But in the manuscript it is the fourth section.

  • Search Methods, Eligibility Criteria and Screening

In my opinion, this section is superfluous, there is no important information on the topic under consideration. It can be replaced with 1-2 sentences (where did the data come from and for how long).

  • Involvement of fALS-associated proteins in mitochondrial dysfunction: an update.

The authors consider in detail the possible mechanisms leading to mitochondrial dysfunction in familial forms of ALS. Proteins are considered, mutations in which lead to the greatest number of cases of fALS.

-page 6, last paragraph, this sentence not clear to me.

«Accounting for over 40% of fALS cases and up to 7% sALS (Mathis et al., 2019), the repeat expansion of the non-coding hexanucleotide GGGGCC (G4C2) in intron 1 of the c9orf72 gene [7,53], originally identified in 2011, is the most frequent genetic mutation associated with ALS. C9ORF72-ALS related mutation».

-page 8, last paragraph

«FUS was reported to partially localize in mitochondria upon mutation or following cellular stress, leading to the increase of FIS1, mitochondrial fragmentation, ROS produc-tion and loss of membrane potential [87,89]». The authors mention FIS1, however, they do not explain what this protein is, its functions and importance in the context of the events under consideration. This needs to be done.

-page 8, last paragraph et seq.

wt replaced by wild type

  • The threshold effect of mitochondrial dysfunction.

- In the last section of the manuscript, the authors dwell on known mitochondrial disorders in sporadic forms of ALS. They are summarized in the table. It would be interesting to summarize and add to this table the data on familial forms of ALS, even though they are partially represented in Figure 1. This comparison is important because familial and sporadic forms are very different from each other.

- This last section should be divided into two parts 1) about sporadic forms 2) the threshold effect of mitochondrial dysfunction (we are talking about all forms of ALS).

     5)  Conclusion is required as a separate section.

Reviewer 2 Report

The past decade witnessed great progress in the mechanistic studies of the ALS associated proteins, especially after the adaptations of novel disease models such as iPSCs derived motor neurons. Thus, the review by Candelise et al is a timely update of these advancements, which will be inspiring to both scientists conducting basic research and MDs doing ALS treatment. I have several recommendations here for the authors to improve the manuscript:   

  1. Both Figure 1 and Table 1 are oversimplified. For Figure 1, the authors may consider: 1. Categorize these ALS-associated proteins based on their sub-mitochondrial localization, such as the matrix, IMM, IMS, OMM, MAM, etc. 2. Categorize these proteins based on the cellular/subcellular localization of the study: neuronal body, axons and synapses, microenvironment of motor neurons (muscle, glia).
  2. The authors denoted that ALS is a non-cell autonomous disease at the beginning of the abstract, yet there lack discussions about the hostile microenvironment under ALS that made iPSCs-based MN or glia transplantation failed once and again. In the last session of the manuscript the authors may elaborate more on mitochondrial defects occurring in the microenvironment (muscle or glia), or how microenvironmental factors can affect mitochondrial abnormalities in motor neurons.

Minor issues:   

Page 4 line 17-22: In the case of ALS, mitochondria are usually thought to have Ca2+ overload issue (rather than a Ca2+ deficiency), which may accelerate the production of ROS or trigger mPTP opening. The authors may need to be more specific on the effect of VDAC inhibition on mitochondrial Ca2+ homeostasis.

Page 2 line 17: “direct data linking…” sounds awkward, it would be better to change to “data directly linking …”

Page 2 line 23 “the collected evidence…” I believe the authors mean “the collective evidence…”   

Page 2 line 27 “on their involvement…” I believe the authors mean “on the studies of their involvement”

Page 2 line 28 “their impairment” or “mitochondrial impairment”?

Page 4 line 1-2: “a little amount” sounds awkward. The authors can simply say “SOD1 is present in both the cytosol and mitochondrial intermembrane space…”.

Page 4 line 30: “evidence reported that…” I believe the authors mean” “studies reported that…”

Page4 line 38: “Outside of VDAC1…” I believe the authors mean “Aside from VDAC1…”

Page4 line 44: ”identified as mutSOD1 interactor…” can be rephrased to “identified TRAF6 as a mutSOD1 interacting protein”.

Page 4 line 44: “multifaceted” or “multifunctional”?

Page 5 line 3: ”Accumulating evidence reported…” can be rephrased to “Accumulating evidence shows”

Page 5 line 8: ”or affecting mitochondrial functionality” I believe the authors mean “or nuclear encoded  but affects mitochondrial functionality”.

Page 5 line 12: “membrane potential” Please specify it is mitochondrial inner membrane potential.

Page 5 line 16: why this reference takes a different format?

Page5 line 15-21: This paragraph does not seem to focus on TDP43 and mitochondrial nucleic acid interactions as described in the first sentence, as the M1, M3 and M5 motifs are not used for RNA binding. The authors can either move the first sentence to the next paragraph or move the content about TDP43 IMM localization to behind next paragraph.

Page 5 line 46-47: “these results indicate that TDP-43 sequesters in condensates the RNA of nuclear-encoded mitochondrial genes, leading to the reduction of their local synthesis.” I believe the authors mean “these results indicate that TDP-43 sequesters the RNA of nuclear-encoded mitochondrial genes in condensates, leading to the reduction of their local translation.”

Page 6 line 36-37: “the association with CTF-25 positive mitochondria” I believe the authors mean “its association with CTF-25 in mitochondria”.

Page 6 line 49: “C9ORF72-ALS related mutation.” I believe this is what the authors forgot to delete.

Page7 line 34: “DRPs” I believe the authors mean “DPRs”

Page 9 line 12: “cytochrome-c oxidase complex IV” please put parenthesis around complex IV as it is the same thing as cytochrome-c oxidase.

Page 10 line 26: Please specify the MtISR mentioned here is in skeletal muscle, or the “secretion of myokines” in the next line sounds confusing.

Page 12 line 17: “IP3R3” or “IP3R3” are both fine. But please keep the format constant throughout the context.

Page 14 line 26: “evidence reported…” Please rephrase to “evidence emerged to support” 

Round 2

Reviewer 2 Report

Page 3 (Line 98): The sentence is incomplete (lacks a subject).

The font of Table 1 can be smaller so as to fit the gene symbols into one row.

In Figure 1 , other interactors: “GPR75” is a typo. Please change to “GRP75”.

Author Response

We thank the Reviewer for uncovering the last minor issues of our manuscript.

The typo in the figure has been corrected, the table width has been increased to fit gene names in one row and the editing mistake in the sentence has been adjusted and labeled in blue.

We hope the Editor and the Reviewer will find our manuscript suitable for publication.
